# Shoulder, Trunk, and Hip Sagittal Plane Kinematics during Stand-Up Paddle Boarding

**DOI:** 10.3390/sports11080152

**Published:** 2023-08-13

**Authors:** Jamie E. Hibbert, Corina Kaufman, Deanna J. Schmidt

**Affiliations:** Department of Kinesiology, California State University San Marcos, San Marcos, CA 92096, USA; ckaufman0296@sdsu.edu (C.K.); djschmidt@csusm.edu (D.J.S.)

**Keywords:** water sports, paddling, recreational sport, motion capture

## Abstract

Stand-up paddle boarding (SUP) as both a competitive and recreational sport has grown in popularity over the last decade. Better understanding paddling kinematics is beneficial for both injury prevention and informing coaching practices in this growing sport. The purpose of this study was to analyze sagittal plane kinematics during both standing and kneeling paddling postures commonly adopted by injury-free, recreational SUP participants. Eighteen recreational SUP participants (seven males/eleven females) were asked to complete a series of paddling tasks on a SUP ergometer in two postures, during which kinematic data were acquired. Sagittal plane kinematic data were analyzed for joint excursion, or range of motion used, while paddling on both sides of the body in each posture. Analysis of variance was used to compare joint excursions across tasks. There were no significant differences in hip or trunk sagittal plant excursion between postures. However, there was significantly greater sagittal plane excursion at the shoulder in the kneeling as compared to the standing posture with the shoulder opposite the paddling side demonstrating the greatest total excursion. These results help establish the parameters of the paddling technique currently in use among injury-free SUP participants and may be used in the future to inform coaching practices.

## 1. Introduction

Stand-up paddle boarding (SUP) is an aquatic sport that has spiked in popularity over the last decade. According to the Outdoor Foundation, there has been an increase of over 1.5 million Americans participating in the sport since 2013 with the total number of participants in the United States estimated at 3.5 million [1]. Previous research has established that individuals who participate in SUP in both competitive and recreational capacities have higher aerobic capacity and anaerobic power than sedentary individuals. Additionally, individuals who participate in SUP also exhibit significantly better balance and isometric endurance than their sedentary peers [2]. Furthermore, the results of a training study conducted with SUP naive participants showed that not only were there significant improvements in balance, aerobic, and anaerobic fitness following SUP training but there were also significant improvements in physical and psychological markers of quality of life [3]. Taken together, it is easy to understand why there has been such exponential growth in SUP participation in the last decade.

Due to the relative simplicity of SUP, most recreational participants are self-taught and are unaware of specific techniques that may help optimize performance and avoid injury. Previous research has reported the injury instance among SUP participants to be 1.95–3.63 injuries per 1000 h of participation [4,5,6]. For comparison to the injury instance in other water sports, the major acute injury instance among competitive surfers and kite surfers is 1.15 and 7 injuries per 1000 h, respectively [7,8]. Injuries to the shoulder and upper arm have been the most common injuries reported among SUP participants [4,5,6,9,10]. Similarly, in studies that have examined the occurrence of pain and injuries among participants engaged in other paddle sports, such as canoeing, kayaking, and whitewater rafting, shoulder pain and upper extremity injuries are highly prevalent [9,10,11]. Lower back injuries also consistently comprise a large percentage of injuries reported across paddle sport participants, including SUP [4,5,6,9,10]. This type of epidemiological information is valuable for understanding the common injury types and prevalence among paddle sport participants, but more research is required to better understand the mechanisms of these injuries as well as factors that may increase injury risk.

To do this, commonly adopted kinematic patterns must first be defined. To this point, there has only been one study published that reported SUP kinematics. The focus of the previously published kinematic study was to establish the difference in stroke kinematics between highly competitive (expert) and novice SUP participants. The results of this study indicated that there are significant differences in the paddling kinematics between these two groups [12]. While the necessity of understanding the kinematic differences between participants at the extreme ends of the SUP experience spectrum is important, these previously characterized groups do not capture the majority of the participants in this fast-growing sport. Further research is necessary to determine how the kinematics of recreational SUP participants compare to the kinematics of these previously characterized groups. Furthermore, in the paddle sports of canoeing and kayaking, an asymmetrical paddle stroke has been associated with injury instance [13,14,15]. This type of asymmetry can be easily identified with kinematic analysis.

Therefore, there is a need to establish the kinematic characteristics of the SUP paddling technique adopted by injury-free recreational participants. These findings may be used in the future to compare kinematic patterns adopted by SUP participants who have sustained injury during recreational SUP or those who chronically experience pain during participation. The differences between these two groups of recreational SUP participants can help to inform paddling technique and improve coaching for the recreational SUP participant. As previously outlined, the multifaceted benefits of SUP participation have been established and thus support the need for additional research that may help better understand the kinematics associated with injury-free participation. Establishing a functional profile for “normal” movement at a joint in a specific sporting population is an important first step in injury prevention. This previously established strategy for, specifically, shoulder injury prevention then uses the “normal” profile as a point of comparison to make adjustments to sport-specific movements in participants learning the sport, experiencing pain, or recovering from injury [16].

The purpose of the current study was to analyze sagittal plane kinematics during both standing and kneeling paddling postures commonly adopted by recreational SUP participants. We hypothesized that there would be greater sagittal plane joint excursion at the trunk and hip while standing, and greater sagittal plane joint excursion at the shoulder while kneeling.

## 2. Materials and Methods

Participants were recruited from the university and the surrounding community. California State University San Marcos is located approximately 40 km from the Pacific Ocean and there are many lakes and inlets in the surrounding area of Southern California that are commonly used by SUP participants. This proximity to areas commonly used for recreational SUP and the rising participation in the sport lent to a broad pool of potential study participants. Participants were recruited using flyers, face-to-face requests, and announcements by faculty in the Department of Kinesiology during their classes and on their course websites. G*Power Analysis statistical software (v. 3.1.9.7, Universitat Keil, Germany) was used prior to data collection to determine that 14 participants were needed for the study to power repeated measures ANOVA with a medium effect size and level of significance of *p* < 0.05.

Eighteen recreational stand-up paddleboarders (seven males/eleven females) participated in this study after providing written informed consent. The mean values and standard deviation (SD) of participants’ age was 22.8 ± 2.63 years, height was 1.72 ± 0.09 m, and body mass was 67.19 ± 9.95 kg. Prior to completing any study-related activities, all participants completed a questionnaire to ensure that they were healthy and free of any musculoskeletal injury at the time of participation. Testing for each participant was completed during one session lasting approximately 45 min. All procedures were approved by the Institutional Review Board at California State University San Marcos on 20 February 2019.

Kinematic data were captured at 120 Hz using a ten-camera, infrared digital camera system (Qualisys AB, Goteborg, Sweden). Participants were fitted with 32 retroreflective markers to define anatomical landmark locations during testing. Markers were placed on the sternal notch, spinous processes of C7, T8, and L5, bilaterally on the acromion process, medial and lateral epicondyles of the humerus, midpoint of the brachium and antebrachium, radial and ulnar styloid process, anterior superior iliac spine, posterior superior iliac spine, apex of the iliac crest, greater trochanter, midpoint of the femur, and medial and lateral joint line of the knee.

Participants were given time to warm up and familiarize themselves with the stand-up paddleboard ergometer (KayakPro SUPErgo, Miami, FL, USA). During this time, they were prompted to paddle on both sides at a self-selected cadence. After participants were comfortable with the set-up, a static trial with the participant standing in anatomical position was captured. Participants were instructed to remain standing and begin paddling at a comfortable cadence (Figure 1A). Once the participant reached a steady cadence, a 20 s trial was recorded. This procedure was completed three times. Participants were given at least 20 s of rest between each trial. The same procedure was then completed with participants paddling on the opposite side. The order in which the left-side paddling (LSP) and right-side paddling (RSP) were completed was randomized. After both LSP and RSP were completed with the participants standing, the procedures were repeated in a tall kneeling posture (Figure 1B). In this kneeling posture, participants were instructed to adopt hip positioning similar to anatomical position rather than sitting back on their heels and flexing the hips.

Data were analyzed using Visual 3D software (C-Motion, Germantown, MD, USA). Three complete strokes from each trial were analyzed from each trial for each participant. This was done by first creating a biomechanical model using marker placement from the static trial, the participant’s height, and body mass. The model identified where each of the markers were relative to one another while the participant was standing still in anatomical position. Once this relationship was established, the model was used to calculate sagittal plane joint excursions at the trunk and bilaterally at the shoulders and hips by tracking the movement of the markers through space in three dimensions. Specifically, the model translated the change in position of the retroreflective markers to the range of motion (ROM) at each of the joints being measured. ROM was then translated to joint excursion. The joint excursions reported represent the total sagittal plane ROM used and are, therefore, all presented as positive values. In addition to reporting the total excursion at the shoulders, trunk, and hips, we also examined the percentage of available ROM utilized at each joint to better understand the contribution of each segment to the overall propulsion movement. A published clinical reference was used to determine the available ROM at each joint [17].

Statistical analysis was completed using SPSS (IBM, Armonk, NY, USA). Means and standard deviations (SD) were computed for joint excursions and the percentage of ROM utilized. The skewness and kurtosis of the difference scores were assessed to confirm the assumption of normality. Repeated measures analysis of variance (ANOVA) was used to compare joint excursions within subjects for both standing and kneeling paddling. Separate ANOVAs were computed for the LSP condition and the RSP condition. A Greenhouse–Geisser correction was used if Mauchly’s Test of Sphericity indicated data were not spherical. Bonferroni post hoc comparisons were used to determine significant differences among joint excursion measures. Effect size was assessed by calculating partial eta squared. An a priori alpha level for significance was set at 0.05.

## 3. Results

There were no significant differences in trunk or hip excursion when comparing standing and kneeling postures regardless of whether the participant was paddling on the left side (trunk: *p* = 0.34; left hip: *p* = 0.28; right hip: *p* = 0.42) or right side (trunk: *p* = 0.43; left hip: *p* = 0.34; right hip: *p* = 0.07) (Table 1). Therefore, our hypothesis that there would be greater sagittal plane joint excursion at the trunk and hip while standing as compared to kneeling was not supported by the data. Less than 20 degrees of hip or trunk motion was demonstrated in both the standing and kneeling postures for both left and right-side paddling (Table 1).

However, the data do support the second portion of our initial hypothesis that there would be greater sagittal plane joint excursion at the shoulder while kneeling as compared to standing. This was true when comparing kneeling versus standing for both shoulders during LSP (main effect F = 64.56, *p* < 0.001, effect size = 0.792; left shoulder: *p* = 0.02, effect size = 0.831; right shoulder: *p* = 0.01), but only the right shoulder demonstrated greater excursion while kneeling compared to standing during RSP (main effect F = 83.52, *p* < 0.001; left shoulder: *p* = 0.06; right shoulder: *p* = 0.03). Consistently, the shoulder opposite the paddling side had the greater sagittal plane joint excursion in both the standing and kneeling postures, with excursion increasing from 11.0 to 13.1 degrees and 13.7 to 14.8 degrees, respectively (Table 1).

The percentage of the available ROM utilized at each joint for both RSP (Figure 2) and LSP (Figure 3) indicate that participants used a greater proportion of the motion available at the shoulder than the trunk or hips. When paddling on both right and left sides, a larger percentage of available shoulder sagittal plane ROM was utilized by the shoulder on the side contralateral to the paddling in the kneeling versus standing posture. Specifically, during RSP, participants used an average of 46.5 ± 13.9% of the ROM available in their left shoulder when kneeling and 37.7 ± 15.1% when standing. The same pattern held true for LSP with the right shoulder moving through 47.4 ± 15.4% of available ROM when kneeling and 38.7 ± 16.2% when standing. Regardless of the paddling posture, the joint at which participants took the least advantage of the available ROM was the hip. A greater percentage of available ROM was used in both hips during standing as compared to kneeling, regardless of paddling side. When RSP in the standing posture, participants used 11.6 ± 8.1% and 12.4 ± 8.0% of available ROM in the left and right hips, respectively, and when kneeling, 7.2 ± 5.4% and 7.6 ± 6.0% of available ROM was used. A similar pattern was observed during LSP with 13.1 ± 7.7% and 10.8 ± 7.3% of left and right hip available ROM used during standing and 8.2 ± 6.4% and 6.9 ± 5.4% used during kneeling. There was very little difference in the average percentage of available trunk ROM used across conditions; 20.8 ± 6.5% and 20.1 ± 6.0% of available trunk ROM were used during RSP and LSP while standing. These values decreased slightly when participants adopted the kneeling posture to 16.3 ± 6.3% and 15.8 ± 6.5% for RSP and LSP, respectively.

## 4. Discussion

The purpose of the current study was to analyze sagittal plane kinematics during both standing and kneeling SUP paddling in injury-free recreational participants. There has only been one other study published that analyzed the kinematics of the paddling motion used in stand-up paddle boarding. That study compared the kinematics of experienced, highly competitive SUP athletes and inexperienced participants [12]. The previous study also only analyzed the SUP stroke in the standing position, which is the stroke required during competitive SUP [12]. For the current study, we decided to collect data for both standing and kneeling paddling postures because, although standing is the only posture that is allowed during SUP competition, it is not uncommon for recreational SUP participants to adopt other postures. One reason that recreational SUP participants may adopt a kneeling or seated posture is to lower their center of gravity thereby increasing their stability on their board. This can be helpful for participants who may be new to SUP and struggle with maintaining balance or those who are paddling in more turbulent water.

It is difficult to make direct comparisons between the data presented in this study and the previously published kinematic data from Schram and colleagues [12] because of differences in processing and presenting the data. For instance, in the previous study, the joint angles from left and right-side paddling were averaged to give a single value for ROM at each joint that was assessed. Another major difference between these two kinematic studies was that in the previously published work participants were required to maintain a power output of 20 W for a total of 40 s during each trial, while our trial had no requirement for maintaining a specified power output. This requirement makes sense for the previous study because they were comparing participants from very different experience levels. The goal of the study presented here was to examine the sagittal plane kinematics of injury-free SUP participants, thus we decided that requiring a specific power output may result in participants altering the kinematic patterns they typically adopt in their recreational SUP outings. Regardless, the findings presented in the current paper regarding total shoulder joint excursion utilized in both standing and kneeling postures bear similarities to the data presented for the inexperienced group in the previous study. However, the total shoulder joint excursion for the recreational SUP participants in this study was higher than the reported ROM in the experienced group from the previous study [12]. This agrees with the conclusion that inexperienced SUP participants use greater shoulder ROM while paddling than experienced participants do. The joint excursions at the trunk and hip differ between the two studies with less trunk ROM and greater hip ROM reported for both the experienced and inexperienced groups by Schram and colleagues [12]. The reliance on greater shoulder ROM coupled with relatively small hip and trunk ROM used throughout the paddling stroke in the current study may suggest that these recreational SUP participants rely on the strength of their shoulder musculature to propel them while using the trunk and hip musculature to stabilize them as they paddle.

As previously mentioned, in the paddle sports of canoeing and kayaking, asymmetry in paddling technique between sides is associated with injury risk [13,14,15]. The data presented in this study did not identify any statistically significant differences between LSP and RSP. All of the participants in this study were injury-free, so it is not possible to determine from this data set whether asymmetries in the SUP paddling technique are also associated with injury risk. Further investigation is needed to elucidate whether this link between paddling stroke asymmetry and injury risk is also present in the SUP participant population.

In addition to understanding the kinematics of recreational SUP participants, other factors have been linked to increased injury risk among SUP participants. For example, previous studies have indicated that there is a relationship between injury and fewer weekly resistance training sessions [5,9]. Although the specific exercises completed during these resistance training sessions were not discussed in detail, this relationship may indicate that lower levels of strength could put SUP participants at higher risk of injury. Further investigation into the type(s) of training and muscle groups that should be targeted with this training would provide important information for recreational SUP participants who want to decrease the likelihood that they sustain an injury. Further, this information could be beneficial for recreational SUP participants who live in areas where weather constrains participation for portions of the year. They may be able to engage in resistance training during the months they are unable to engage in SUP, which would potentially improve performance and decrease injury risk once they are able to be on the water again.

Moreover, greater total training volume has been shown to be associated with increased injury risk [5]. Therefore, the level of participation is a factor when assessing the risk of shoulder injury as competitive participation would likely increase training volume and thus increase overall frequency and paddling duration as compared to a recreational level of participation [16,18]. However, more data are needed to determine whether this link between training volume and injury risk persists across the spectrum of recreational SUP participants.

Previous studies have reported the relative risk of injury while participating in SUP to be between 1.95 and 3.63 injuries per 1000 h of participation. This injury risk differed depending on the level of participation with elite participants having a higher relative risk than recreational participants [5,6]. Investigations into injury prevalence in SUP participants have reported that 26–46% of injuries sustained by participants are to the shoulder [4,6,9,10]. Another study that grouped arm and upper thoracic injuries together reported an even higher injury prevalence of 59% [5]. The data presented in this paper showing that a higher percentage of available ROM at the shoulder is used than at any other joint may help to explain why the most prevalent injuries reported in paddle sports, more specifically SUP, are to the shoulder/upper arm. It is well understood that the shoulder joint exhibits a trade-off between stability and mobility. As more of the range of motion available at the joint is used, i.e., the joint moves into the extreme positions of flexion and extension, the joint becomes less stable due to less contact between the head of the humerus and the glenoid fossa. While placing the joint in a less stable position does not immediately result in injury, the risk of injury does increase with prolonged time spent in these positions, especially if that increased time is coupled with any perturbation [19]. While it is not possible to state whether the recreational SUP participants in this study are at higher risk of injury than the elite participants categorized in the previous kinematic study, further research is needed to gather kinematic data from recreational SUP participants who experience pain while paddling or have sustained a shoulder injury. These data would help establish whether or not a link between joint excursion and injury risk in recreational SUP participants is present.

Taken together, the risk of injury while participating in SUP is low, but if an injury is sustained, it is likely to occur at the shoulder. One suggestion that may help to decrease injury risk in recreational SUP participants would be to spend the majority of their paddling sessions standing rather than kneeling to decrease the length of time that they may be using a very large proportion of the ROM available at their shoulder joints. However, further investigation into the link between increased joint excursion and injury risk is needed.

One limitation of this study is that it was performed with the participants paddling on a SUP ergometer in the laboratory. Utilizing an indoor setting allowed for better kinematic data collection than would be possible on water, but the lack of variables introduced by wind and water conditions may have altered the paddling motion adopted by participants. It should be noted that the performance of participants using this SUP ergometer is highly correlated to their performance over water, but there have not been any studies conducted that compare the kinematics while paddling in open water with those employed while using a SUP ergometer [20].

In conclusion, the data presented in this study are some of the first kinematic data collected on healthy, recreational SUP participants. The primary objective of this study was to present these data so that they may be used to better understand the kinematics typically adopted by injury-free recreational SUP participants. There are some key differences between competitive and recreational SUP paddling movements. One key difference is the adoption of a kneeling posture which is not allowed during competition. Better understanding the movement patterns of healthy participants is necessary for improving instruction for beginning SUP participants to encourage them to adopt kinematics that will help them remain injury-free and able to participate in this growing sport.

## Figures and Tables

**Figure 1 sports-11-00152-f001:**
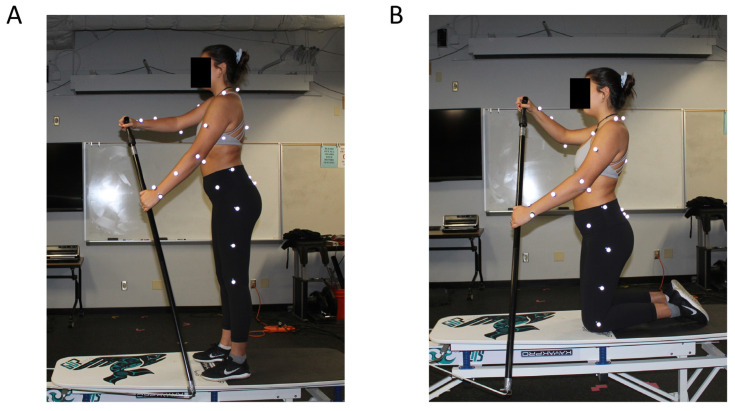
Experimental set-up for (**A**) standing and (**B**) kneeling paddling postures.

**Figure 2 sports-11-00152-f002:**
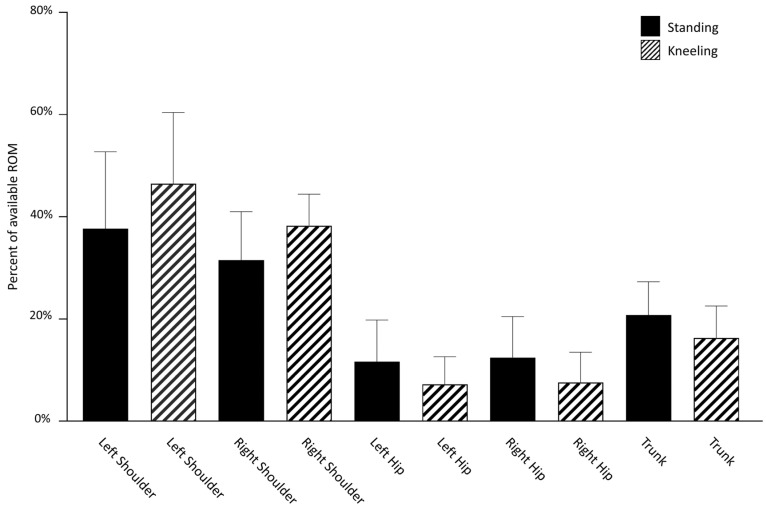
Right-side paddling: range of motion used at each joint as a percentage of the total range of motion available at the joint.

**Figure 3 sports-11-00152-f003:**
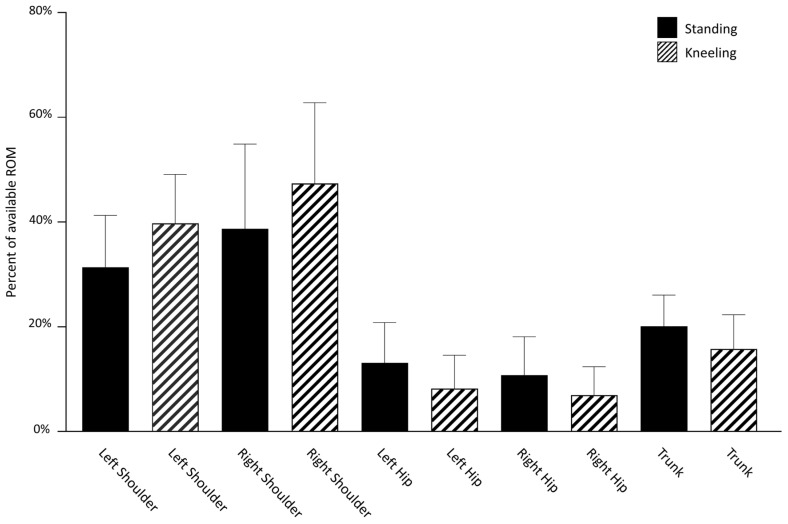
Left-side paddling: range of motion used at each joint as a percentage of the total range of motion available at the joint.

**Table 1 sports-11-00152-t001:** Mean excursion at each joint in all paddling conditions. The mean sagittal plane joint excursion in degrees is expressed as mean ± SD. Excursion is listed for each joint measured and categorized by posture and paddling side.

	Standing		Kneeling	
	Right-Side Paddling	Left-Side Paddling	Right-Side Paddling	Left-Side Paddling
Shoulder				
Left	67.79 ± 27.09	56.48 ± 17.85	83.64 ± 25.08	71.57 ± 16.84 *
Right	56.72 ± 17.06	69.64 ± 29.17	68.84 ± 11.11 *	85.25 ± 27.77 *
Trunk	16.62 ± 5.22	16.10 ± 4.77	13.04 ± 5.00	12.61 ± 5.22
Hip				
Left	17.48 ± 12.22	19.68 ± 11.55	10.84 ± 8.09	12.33 ± 9.54
Right	18.66 ± 12.04	16.14 ± 10.99	11.33 ± 8.93	10.40 ± 8.15

Asterisk (*) indicates a significantly greater excursion value compared to standing, *p* < 0.05.

## Data Availability

The data presented in this study are available upon request. Please contact the corresponding author with such requests.

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
