# Peer review of "Shoulder, Trunk, and Hip Sagittal Plane Kinematics during Stand-Up Paddle Boarding"

_sports, 2023, doi:10.3390/sports11080152_

Round 1
Reviewer 1 Report
The paper is well written. However, I have a few comments that need to be corrected. Abstract - This is the first item I read. You need to clarify what do you understand by joint excursion? Is it maximum angular values or range of motion? I understand range of motion. This information is in line 95. It would be better to write in the abstract range of motion instead joint excursion.
Lines 90 – 99 – how did you calculated ROM?
Lines 96 – 99 – Please describe how "available ROM" was assessed at each joint?
Lines 100 – 108 – Was the normality of the distribution tested? With what test?
Table 1 – Please improve the readability of the table. The table can be expanded. Unfortunately, the individual elements overlap. I don't understand why a part is bold. Please title the table better. What is under the table can be moved above and then the title will not be questionable.
What is the difference between figures 2 and 3? For me, only figure 3 is visible - twice. I think this is an error. Please correct it. It seems to me that instead of figures, a table would be better.
Author Response
Reviewer 1 Comments for authors
Reviewer comment: The paper is well written. However, I have a few comments that need to be corrected. Abstract - This is the first item I read. You need to clarify what do you understand by joint excursion? Is it maximum angular values or range of motion? I understand range of motion. This information is in line 95. It would be better to write in the abstract range of motion instead joint excursion.
Response: Thank you for bringing this ambiguity to our attention. We have added the following language to the abstract to clarify what is meant by the term ‘joint excursion.’ Sagittal plane kinematic data were analyzed for joint excursion, or range of motion used, while paddling on both sides of the body in each posture.
Additionally, we have also made sure that the term ‘excursion’ is clear in the methods to avoid any ambiguity.
Reviewer comment: Lines 90 – 99 – how did you calculated ROM?
Response: The range of motion is calculated using the Visual 3D software that is referenced in the text. In order to clarify how this was done and the process used by the program the following language has been added. The model identified where each of the markers are relative to one another while the participant was standing still in anatomical position. Once this relationship was established, the model was used to calculate sagittal plane joint excursions at the trunk and bilaterally at the shoulder and hip by tracking the movement of the markers through space in three dimensions. Specifically, the model translated the change in position of the retroreflective markers to the range of motion (ROM) at each of the joints being measured.
Reviewer comment: Lines 96 – 99 – Please describe how "available ROM" was assessed at each joint?
Response: The reference that was used to determine the available ROM is a very commonly used clinical reference that states the average available ROM for every joint. The following language has been added to clarify that the reference cited on line 105 is the reference that we used A published clinical reference was used to determine the available ROM at each joint [10].
Reviewer comment: Lines 100 – 108 – Was the normality of the distribution tested? With what test?
Response: The assumption of normality was tested by using SPSS statistical software to evaluate skewness and kurtosis of the difference scores for data used for repeated measures ANOVA for both right side paddling and left side paddling. Skewness and kurtosis values were below 2 indicating normality of the data distribution. Therefore, the assumption of normality was met and the data were appropriate for ANOVA statistical comparisons.
Methods now state: The skewness and kurtosis of the difference scores were assessed to confirm the assumption of normality.
Reviewer comment: Table 1 – Please improve the readability of the table. The table can be expanded. Unfortunately, the individual elements overlap. I don't understand why a part is bold. Please title the table better. What is under the table can be moved above and then the title will not be questionable.
Response: We thank the reviewer for this comment and are concerned about the problems indicated in this comment. Unfortunately, in the version of the manuscript that we were provided for revisions we do not see the overlap or differences in bold font described by the reviewer. We are confident that the journal editorial staff will confirm that the table displays correctly prior to publication. We have changed the table title to Mean excursion at each joint in all paddling conditions.
We have also moved most of the text that was below the table to above the table, as suggested.
Reviewer comment: What is the difference between figures 2 and 3? For me, only figure 3 is visible - twice. I think this is an error. Please correct it. It seems to me that instead of figures, a table would be better.
Response: We thank the reviewer for identifying that figure 2 was mislabeled and there was an error in the figure legend. Figure 2 shows the percent of available ROM used at each joint in standing and kneeling postures when the participants were paddling on the right side of the body. Figure 3 shows percent of available ROM used at each joint in standing and kneeling postures when the participants were paddling on the left side of the body. We thank the reviewer for the suggestion of presenting these data in a table, however, we feel that the ability to visualize the pattern in the percent of ROM used and how that changes between kneeling and standing is very valuable. Visualizing these patterns is more easily accomplished using bar graph figures rather than a table. To add clarity, we also reworded the figure captions to better show that Figure 2 is right side paddling and Figure 3 is left side paddling.
Reviewer 2 Report
For authors
Introduction
- It would be useful for the reader to understand the limitations or criticisms of previous studies that your work aims to address.
- When discussing the instance of injuries, it would be beneficial to provide a comparison with similar sports to give readers a better understanding of the risk level in SUP.
- Expanding your review of related studies, even if they are not specifically focused on SUP, could strengthen your argument about the need for more research in this area.
- Make a stronger link between your study objectives and the problems identified in the field.
Method
- It may be useful to describe how these participants were recruited and why this specific group was chosen.
- Also, provide sample size calculations for statistical power.
Results
- Nothing to report.
Discussion
- The authors compare their findings with those of a previous study by Schram et al. (2019) [review citation style] and mention that it is challenging to make direct comparisons due to differences in data processing and presentation. Describe those diferences.
- The authors mention that recreational SUP participants sometimes adopt a kneeling or seated position to lower their center of gravity, thereby increasing their stability. However, they do not provide any quantitative data on how often participants switch between standing and kneeling or seated positions.
- The authors mention that the most common injuries reported in SUP are to the shoulder/upper arm, which they attribute to the high ROM. However, they do not discuss whether this high ROM is inherently harmful or whether it might be due to other factors, such as poor paddling technique or lack of shoulder strength and stability.
- The authors suggest that standing rather than kneeling during paddling might decrease the injury risk by reducing shoulder ROM. However, this recommendation is not backed up with clear evidence from the study.
- The authors acknowledge the limitation of conducting the study on a SUP ergometer in a laboratory, lacking real-life paddling conditions. While they mention that ergometer performance is highly correlated with performance on water, it might be useful to clarify that this correlation applies to performance outcomes rather than biomechanical paddling patterns.
- The authors mainly focus on sagittal plane kinematics in relation to injury risk. However, they could also discuss other factors that might contribute to injury risk in SUP, such as the frequency, duration of sessions, and paddling technique.
Minor editing of English language required
Author Response
Reviewer 2 Comments for authors
Introduction
Reviewer comment: It would be useful for the reader to understand the limitations or criticisms of previous studies that your work aims to address.
Response: The following language has been added/ amended to further explain some of the limitations of previous work on SUP: To this point, there has only been a single study published that has reported on SUP kinematics. The focus of the previously published kinematic study was to establish the difference in stroke kinematics between highly competitive SUP participants and novice SUP participants [12]. While the necessity of understanding the kinematic differences between these participants at the extreme ends of the SUP experience spectrum is important, these groups do not capture the majority of the participants in this fast-growing sport.
Despite the rate at which this sport is growing, there have been very few research studies published on the sport in general, fewer still on the kinematics of the stroke, and none characterizing the paddling kinematics adopted by recreational SUP participants. Thus, there is a need to establish the kinematic characteristics of SUP paddling technique adopted by injury-free recreational participants. These findings may be used in the future to compare kinematic patterns adopted by SUP participants who have sustained injury during recreational SUP or those who chronically experience pain during participation. The differences between these two groups of recreational SUP participants can help to inform paddling technique and improve coaching for the recreational SUP participant. As previously outlined, the multifaceted benefits of SUP participation have been established which support the need for additional research that may help better understand kinematics associated with injury-free participation. This strategy of functionally profiling the “normal” joint of interest in a specific sporting population and using this profile to identify the difference between this profile and that of the same joint in injured participants of the same sport has been a previously established method for preventing injury, specifically shoulder injury, in sport [13].
Reviewer comment: When discussing the instance of injuries, it would be beneficial to provide a comparison with similar sports to give readers a better understanding of the risk level in SUP.
Response: We have expanded this paragraph in the introduction to include information about injury instance in other water sports and injury epidemiology in other paddle sports to contextualize the injury risk of SUP participation and contextualize the types of injuries commonly sustained during paddle sport participation. The text now reads: Previous research has reported the injury instance to be 1.95 – 3.63 injuries per 1000 hours of participation [1-3]. For comparison to the injury instance in other water sports, major acute injury instance among competitive surfers and kite surfers is 1.15 and 7 injuries per 1000 hours, respectively [4, 5]. Injuries to the shoulder and upper arm have been reported to be the most common among SUP participants [1-3, 6, 7]. In studies that have examined the occurrence of pain and injuries among participants engaged in other paddle sports, such as canoeing, kayaking, and whitewater rafting, shoulder pain and upper extremity injuries are highly prevalent [6-8]. Lower back injuries also repeatedly comprise a large percentage of injuries reported across paddle sport participants, including SUP [1-3, 6, 7].
Reviewer comment: Expanding your review of related studies, even if they are not specifically focused on SUP, could strengthen your argument about the need for more research in this area.
Response: We have expanded this paragraph in the introduction to include information about injury instance in other water sports and injury epidemiology in other paddle sports. The text now reads: Previous research has reported the injury instance among SUP participants to be 1.95 – 3.63 injuries per 1000 hours of participation [4-6]. For comparison to the injury instance in other water sports, major acute injury instance among competitive surfers and kite surfers is 1.15 and 7 injuries per 1000 hours, respectively [7, 8]. Injuries to the shoulder and upper arm have been reported to be the most common among SUP participants [4-6, 9, 10]. Similarly, in studies that have examined the occurrence of pain and injuries among participants engaged in other paddle sports, such as canoeing, kayaking, and whitewater rafting, shoulder pain and upper extremity injuries are highly prevalent [9-11]. Lower back injuries also repeatedly comprise a large percentage of injuries reported across paddle sport participants, including SUP [4-6, 9, 10]. This type of epidemiological information is valuable for understanding the common injury types and prevalence among paddle sport participants, but more research is required to better understand the mechanisms of these injuries as well as factors that may increase injury risk.
We have also added more detail about the findings of the previously published work to more clearly articulate what the focus of these studies have been and where there is a gap in knowledge. The following sentence has been added to the paragraph that follows the one quoted in the previous portion of the response to this comment: The focus of the previously published kinematic study was to establish the difference in stroke kinematics between highly competitive SUP participants and novice SUP participants. The results of this study indicated that there are significant differences in the paddling kinematics between these two groups [12]. While the necessity of understanding the kinematic differences between participants at the extreme ends of the SUP experience spectrum is important, these previously characterized groups do not capture the majority of the participants in this fast-growing sport. Further research is necessary to determine how the kinematics of recreational SUP participants compare to the kinematics of these previously characterized groups. Furthermore, in the paddle sports of canoeing and kayaking an asymmetrical paddle stroke has been associated with injury instance [13-15]. This type of asymmetry can be easily identified with kinematic analysis.
Reviewer comment: Make a stronger link between your study objectives and the problems identified in the field.
Response: The following section has been amended to clarify the link between the study objectives and the gap in knowledge identified in the field: Therefore, there is a need to establish the kinematic characteristics of SUP paddling technique adopted by injury-free recreational participants. These findings may be used in the future to compare kinematic patterns adopted by SUP participants who have sustained injury during recreational SUP or those who chronically experience pain during participation. The differences between these two groups of recreational SUP participants can help to inform paddling technique and improve coaching for the recreational SUP participant. As previously outlined, the multifaceted benefits of SUP participation have been established and thus support the need for establishing a functional profile for “normal” movement at a joint in a specific sporting population is an important first step in injury prevention. This previously established strategy for, specifically, shoulder injury prevention then uses the “normal” profile as a point of comparison to make adjustments to sport-specific movements in participants learning the sport, experiencing pain, or recovering from injury [16]. additional research that may help better understand kinematics associated with injury-free participation.
The purpose of the current study was to analyze sagittal plane kinematics during both standing and kneeling paddling postures commonly adopted by recreational SUP participants. We hypothesized that there would be greater sagittal plane joint excursion at the trunk and hip while standing, and greater sagittal plane joint excursion at the shoulder while kneeling.
Method
Reviewer comment: It may be useful to describe how these participants were recruited and why this specific group was chosen.
Response: We thank the reviewer for this comment and have added the following language to the Materials and Methods section: Participants were recruited from the university and the surrounding community. California State University San Marcos is located approximately 40 kilometers from the Pacific Ocean and there are many lakes and inlets in the surrounding area of Southern California that are commonly used by SUP participants. This proximity to areas commonly used for recreational SUP and the rising participation in the sport lent to a broad pool of potential study participants. Participants were recruited using flyers, face-to-face requests, and announcements by faculty in the Department of Kinesiology during their classes and on their course websites.
Reviewer comment: Also, provide sample size calculations for statistical power.
Response: We appreciate the reviewer’s comment and have now added a sentence in the methods detailing how we determined the sample size necessary to achieve statistical power for the study. We have included the following statement in the methods: G*Power Analysis statistical software (v. 3.1.9.7, Universitat Keil, Germany) was used prior to data collection to determine that 14 participants were needed for the study to power repeated measures ANOVA with a medium effect size and level of significance of p < 0.05.
Results
Reviewer comment: Nothing to report.
Discussion
Reviewer comment: The authors compare their findings with those of a previous study by Schram et al. (2019) [review citation style] and mention that it is challenging to make direct comparisons due to differences in data processing and presentation. Describe those diferences.
Response: We thank the reviewer for this comment. We have removed the parenthetical date from this section to avoid confusion. It was included to specify which paper by Schram and colleagues was being referenced, but understand how this may have been distracting.
We have added in the additional information to illustrate the differences between the current study and the previously published kinematic study: It is difficult to make direct comparisons between the data presented in this study and the previously published kinematic data from Schram and colleagues [12] because of differences in processing and presenting the data. For instance, in the previous study the joint angles from left and right-side paddling were averaged to give a single value for ROM at each joint that was assessed. Another major difference between these two kinematic studies was that in the previously published work participants were required to maintain a power output of 20 W for a total of 40 seconds during each trial, while our trial had no requirement for maintaining a specified power output. This requirement makes sense for the previous study because they were comparing participants from very different experience levels. The goal of the study presented here was to examine the sagittal plane kinematics of injury-free SUP participants, thus we decided that requiring a specific power output may result in participants altering the kinematic patterns they typically adopt in their recreational SUP outings. Regardless, the findings presented in the current paper regarding total shoulder ROM utilized in both standing and kneeling postures bear similarities to the data presented for the inexperienced group in the previous study. However, total shoulder ROM for the recreational SUP participants in this study was higher than the reported ROM in the experienced group from the previous study [12].
Reviewer comment: The authors mention that recreational SUP participants sometimes adopt a kneeling or seated position to lower their center of gravity, thereby increasing their stability. However, they do not provide any quantitative data on how often participants switch between standing and kneeling or seated positions.
Response: We thank the reviewer for this thoughtful comment and agree that this information could provide some contextual information. Unfortunately, we did not ask our study participants about this and have not been able to find a source that provides this kind of information. Therefore, we were not able to add quantitative data on this comparison to the manuscript.
Reviewer comment: The authors mention that the most common injuries reported in SUP are to the shoulder/upper arm, which they attribute to the high ROM. However, they do not discuss whether this high ROM is inherently harmful or whether it might be due to other factors, such as poor paddling technique or lack of shoulder strength and stability.
Response: We thank the reviewer for pointing out this opportunity to clarify how joint position and injury risk, especially in the shoulder are linked. The text in this section has been changed to read: It is well understood that the shoulder joint exhibits a trade-off between stability and mobility. As more of the range of motion available at the joint is used, i.e. the joint moves into the extreme positions of flexion and extension, the joint becomes less stable due to less contact between the head of the humerus and the glenoid fossa. While placing the joint in a less stable position does not immediately result in injury, the risk of injury does increase with prolonged time spent in these positions, especially if that increased time is coupled with any perturbation [18]. While it is not possible to state whether the recreational SUP participants in this study are at higher risk of injury than the elite participants categorized in the previous kinematic study, further research is needed to gather kinematic from recreational SUP participants who experience pain while paddling or have sustained a shoulder injury. These data would help establish whether or not a link between joint excursion and injury risk in recreational SUP participants is present.
Reviewer comment: The authors suggest that standing rather than kneeling during paddling might decrease the injury risk by reducing shoulder ROM. However, this recommendation is not backed up with clear evidence from the study.
Response: This point in the discussion was intended to be posed as a strategy that could be taken into consideration. In order to avoid this point being overinterpreted, we have added the following sentence: However, further investigation into the link between increased joint excursion and injury risk is needed.
Reviewer comment: The authors acknowledge the limitation of conducting the study on a SUP ergometer in a laboratory, lacking real-life paddling conditions. While they mention that ergometer performance is highly correlated with performance on water, it might be useful to clarify that this correlation applies to performance outcomes rather than biomechanical paddling patterns.
Response: We thank the reviewer for raising this point and have changed this sentence to read: “It should be noted that the performance of participants using this SUP ergometer is highly correlated to their performance over water, but there have not been any studies conducted that compare the kinematics while paddling in open water with those employed while using a SUP ergometer.”
Reviewer comment: The authors mainly focus on sagittal plane kinematics in relation to injury risk. However, they could also discuss other factors that might contribute to injury risk in SUP, such as the frequency, duration of sessions, and paddling technique.
Response: We thank this comment and have added the following language to the discussion:
As previously mentioned, in the paddle sports of canoeing and kayaking asymmetry in paddling technique between sides is associated with injury risk [13-15]. The data presented in this study did not identify any statistically significant differences between LSP and RSP. All of the participants in this study were injury-free, so it is not possible to determine from this data set whether asymmetries in the SUP paddling technique are also associated with injury risk. Further investigation is needed to elucidate whether this link between paddling stroke asymmetry and injury risk is also present in the SUP participant population.
In addition to understanding the kinematics of recreational SUP participants, there are other factors that have been linked to increased injury risk among SUP participants. For example, previous studies have indicated that there is a relationship between injury and fewer weekly resistance training sessions [5, 9]. Although, the specific exercises completed during these resistance training sessions were not discussed in detail, this relationship may indicate that lower levels of strength could put SUP participants at higher risk of injury. Further investigation into the type(s) of training and muscle groups that should be targeted with this training would provide important information for recreational SUP participants who want to decrease the likelihood that they sustain an injury. Further, this information could be beneficial for recreational SUP participants who live in areas where weather constrains participation for portions of the year. They may be able to engage in resistance training during the months they are unable to engage in SUP that would potentially improve performance and decrease injury risk once they are able to be on the water again.
Moreover, greater total training volume has been shown to be associated with increased injury risk [5]. Therefore, the level of participation is a factor when assessing the risk of shoulder injury as competitive participation would likely increase training volume and thus increase overall frequency and paddling duration as compared to a recreational level of participation [16, 19]. However, more data is needed to determine whether this link between training volume and injury risk persists across the spectrum of recreational SUP participants.